# GeoRDe: tertiary structure-based RNA design with multiple geometric constraint

## Abstract

Functional RNA sequence design plays an essential role in the regulation of life processes. The RNA inverse folding problem, which involves designing nucleic acid sequences based on their three-dimensional structures, remains highly challenging. This complexity arises not only from the inherent flexibility of RNA structures but also from the base-pairing rules that impose critical spatial constraints on the RNA scaffold. In recent times, the design of RNA has often depended on geometric graph networks to design sequences. Motivated by recent advancements in protein design, we have developed the RNAformer module. This module is capable of learning the geometric constraints of RNA molecules in cooperation with geometric graph networks. Furthermore, to enhance the specificity of sequence generation, we have integrated secondary structure information as labels, ensuring that the designed sequences align more closely with secondary structure constraints. Additionally, we have used RNA language models to understand average evolutionary constraints. By incorporating a range of constraint insights, GeoRDe has demonstrated superior performance under identical training data conditions and has also showcased generalization capabilities on the independent casp15 and RNA-puzzle datasets. Through extensive experimentation, the GeoRDe has proven to be an innovative solution to the challenges of RNA design.

## 1 Introduction

Ribonucleic acid (RNA) performs a variety of essential functions within the cell, including but not limited to catalyzing biochemical reactionsLewin (1982), regulating gene expressionPrasanth et al. (2005), and forming components of cellular machineryJinek & Doudna (2009). These multifaceted roles of RNA make it a key target for biomedical research and therapeutic development. Therefore, the design of RNA sequences is not only crucial for understanding its biological functions but also holds significant potential for developing new therapeutic strategiesYin & Rogge (2019)Lu & Thum (2019).

While the diversity and functionality of RNA are largely determined by its three-dimensional (3D) structure, the challenge of inferring the corresponding one-dimensional (1D) sequence from a given 3D structure, known as the RNA inverse folding problemChurkin et al. (2018), remains a significant obstacle. Traditional approaches to this problem have often focused on RNA's secondary structureSzabat et al. (2020)Sato & Hamada (2023). However, with an improved understanding of RNA's 3D geometryTownshend et al. (2021), researchers have begun to explore computational methods that can design RNA sequences directly from its 3D structure. For instance, the RiboDiffusion model uses a generative diffusion model to iteratively transform random sequences into target sequences, thereby learning the conditional distribution of RNA sequences within a specified 3D scaffold structureHuang et al. (2024). Additionally, methods like gRNAdeJoshi et al. (2024) and RDesignTan et al. (2024) encode the 3D framework of RNA using multi-state graph neural networks, providing innovative pathways for RNA sequence design. The success of these methods highlights the potential of deep learning in managing RNA structural data.

Advances in protein structure predictionJumper et al. (2021) and designRen et al. (2024a) have also informed new models and strategies for RNA design. For example, CarbonNovoRen et al. (2024b), which generates protein structures and sequences concurrently through a unified energy-

based model, has shown its effectiveness in protein design. However, the unique features of RNA molecules, such as the specificity of base pairing and the flexibility of the RNA backbone, require that design methods be able to accurately address these distinctive structural characteristics. Moreover, compared to the extensive database of protein structures available in the Protein Data Bank (PDB)Bank (1971), the scarcity of RNA structural data necessitates that design methods exhibit greater data efficiency and generalization capabilities.

Building on the foundation of current research, this study introduces a novel inverse folding algorithm known as GeoRDe (GEOMETRIC CONSTRAINT RNA DESIGN). This algorithm employs innovative approaches to handle the distinctiveness of RNA's three-dimensional structure. Firstly, GeoRDe employs a hybrid architecture that integrates geometric graph networks with triangle attention networks to derive representations of RNA molecules. The triangle attention network represents an advancement over the traditional attention mechanism, specifically adapted to capture the intricate spatial configurations characteristic of RNA molecules. In contrast to geometric graph networks for unstructured data, triangle attention networks excel at capturing proximal bases linked by covalent bonds within structured RNA information. Secondly, the algorithm employs multi-task learning to account for the significance of the base-pairing principle in RNA design. Lastly, the algorithm harnesses large models to extract constraints from evolutionary information.

Our main contributions are summarized as follows:

1. **Innovative Design of the RNAformer Module**: This research has crafted an RNAformer module that collaborates with geometric graph networks to learn the geometric constraints of RNA molecules.

2. **Introduction of Secondary Structure Constraints**: By integrating secondary structure constraints in a labeled format, the precision of the designed sequences is markedly improved.

3. **Embedding of RNA Language Models**: The embedding of RNA language models introduces average evolutionary constraint information for sequences, thereby enhancing their evolutionary reliability.

4. **Validation across Multiple Datasets**: The performance of the algorithm has been validated across various datasets, and the findings indicate that the incorporation of diverse constraint strategies effectively confines the sequence design space and exhibits robust generalization capabilities.

## 2 RELATED WORK

### 2.1 PROTEIN DESIGN

Protein sequence design generally refers to the process of creating amino acid sequences for proteins with specified functions based on requirementsWu et al. (2021)Anand et al. (2022). Since the three-dimensional (3D) structure of a protein largely determines its function, designing sequences based on the protein's 3D structure is a commonly used approach. Recently, methods like Protein-MPNNDauparas et al. (2022) have demonstrated high recovery rates in protein sequence design. ProteinMPNN utilizes deep learning frameworks and message-passing neural networks (MPNN) to achieve this. Integrating pre-trained models with sequence and structural data can provide additional evolutionary information for generating sequences with designated functions. Examples of such methods include protgenFerruz et al. (2022) and ESM3Hayes et al. (2024).

### 2.2 RNA SEQUENCE DESIGN

In recent years, an increasing number of studies have focused on designing RNA sequences to regulate life processes.Isaacs et al. (2006)Peters et al. (2015) RNA sequence design efforts include both approaches that are based on existing RNA sequences and those that focus on RNA structures. DeepCRISPRChuai et al. (2018) integrates unlabeled single-guide RNA (sgRNA) sequences and employs a deep convolutional denoising neural network (DCDNN)-based autoencoder for unsupervised learning. This is complemented by a convolutional neural network trained on labeled sgRNA sequences to facilitate the design of CRISPR guide RNAs. RfamGenSumi et al. (2024) is developed

by training on the Rfam family sequences, utilizing a variational autoencoder (VAE) and a covariance model (CM) to generate synthetic RNA family sequences. EvoNguyen et al. (2024) represents a foundational model for nucleic acids that harnesses deep learning and extensive genomic datasets to systematically engineer RNA sequences tailored for specific functions.

RNA inverse folding is the process of generating one-dimensional RNA sequences based on their secondary or tertiary structures. Techniques such as RNAiFoldGarcia-Martin et al. (2013) employ constraint programming to optimize RNA sequence design to meet specific secondary structure criteria. RNAinverseHofacker et al. (1994) utilizes an adaptive random walk approach, predicting RNA sequences for target structures through iterative mutation and energy minimization. NUPACKZadeh et al. (2011) employs a collective defect optimization strategy to craft RNA sequences that minimize undesirable pairing. RDESIGNTan et al. (2024) leverages a hierarchical data-efficient representation learning framework, integrating cluster-level and sample-level contrastive learning to enhance the design of RNA tertiary structures. gRNAdeJoshi et al. (2024) employs a multi-state graph neural network to generate candidate RNA sequences conditioned on one or more 3D backbone structures, taking into account both RNA structure and dynamics. RiboDiffusionHuang et al. (2024) applies a generative diffusion model to RNA inverse folding design by learning the conditional distribution given 3D backbone structures.

### 2.3 RNA STRUCTURE PREDICTION

RNA structure prediction involves predicting the folding conformation of RNA from its one-dimensional sequence. Initially, RNA prediction efforts concentrated on predicting RNA secondary structures. ViennaRNALorenz et al. (2011)Hofacker (2003) is a physics-based prediction tool that employs a standard energy function to predict RNA secondary structures. SpotRNAYang et al. (2014) introduces a deep contextual learning approach, trained via transfer learning to predict the secondary structure of all base pairs, including atypical and non-nested (pseudoknot) pairs. KnotFoldGong et al. (2024) is an advanced method for accurately predicting RNA secondary structures, including pseudoknots, by integrating learned potentials with minimum-cost flow algorithms and enhancing prediction accuracy through attention-based neural networks. Unlike proteins, RNA three-dimensional structures exhibit greater flexibility. In recent years, with advancements in machine learning, approaches such as AlphaFold3Abramson et al. (2024), RosettaFoldNABaek et al. (2024), trRosettaRNAWang et al. (2023), and RhoFoldShen et al. (2022) have emerged. These methods utilize multiple sequence alignments as input and leverage deep learning networks, often incorporating modules like evofold, to predict the three-dimensional coordinates of RNA.

## 3 METHODS

### 3.1 INVERSE FOLDING PROBLEM DEFINITION

In this paper, RNA inverse folding specifically denotes the process of identifying or engineering RNA sequences capable of folding into a predetermined target structure. For a one-dimensional RNA sequence $S$ comprising $N$ nucleotides, each nucleotide is composed of one of four types of ribonucleotides, denoted as $S \in \{A, U, C, G\}^N$. The secondary structure of RNA is depicted using dot-bracket notation, where the majority of RNA secondary structure pairings are categorized into three types: A-U, C-G, and G-U. Bases adhering to these pairings are denoted by brackets, while those not conforming are indicated by dots.

Regarding the three-dimensional structure of RNA, this paper employs a coarse-grained backbone representation to delineate the 3D configuration. This representation utilizes the C4', C1', N1 atoms to signify pyrimidine nucleotides and the C4', C1', N9 atoms to signify purine nucleotides. The model presented in this paper simulates the conditional distribution of RNA sequences given the three-dimensional structure, which is mathematically represented as $p(S|x)$.

### 3.2 FEATURE REPRESENTATION

The input features in this paper are categorized into two main components. Initially, the coarse spatial arrangement of the RNA backbone is delineated through the local orientation of the C1' atoms. All atoms within a 12 Å radius from each atom are enumerated, and their relative contact

distances are harnessed as pair features to enhance the characterization of the local environment surrounding each atom. Subsequently, to more accurately depict the arrangement of the RNA backbone, a graph-based approach is employed. The unit vectors, distances, angles, and torsion angles of adjacent atoms are extracted as graph node attributes. Adjacent edges in the graph network are defined between atoms that are in close proximity to one another. This methodology more effectively encapsulates the spatial positional information between RNA backbones.

## 3.3 EVALUATION METRIC

In the field of computational RNA design, a series of metrics are commonly used to assess the effectiveness of designed sequences, quantifying the fidelity and structural compatibility of the sequence compared to the target scaffold structure. Here, we evaluate the reliability of the generated sequences from one or three dimensions.

### 3.3.1 NATIVE SEQUENCE RECOVERY

This metric measures the percentage of nucleotides in the designed sequence that accurately recover the native sequence, serving as a direct measure of sequence conservation. The sequence recovery rate is given by:

$$\text{Recovery} = \frac{N_{\text{rec}}}{N_{\text{nat}}} \times 100\% \tag{1}$$

where $N_{\text{rec}}$ is the number of nucleotides accurately recovered in the designed sequence, and $N_{\text{nat}}$ is the total number of nucleotides in the native sequence.

### 3.3.2 MACRO F1 SCORE

The Macro-F1 score is a comprehensive performance metric used to evaluate the accuracy of models in the RNA design task across different classes of RNA letters (A, U, C, G). It is calculated by averaging the F1 scores for each class, where the F1 score for a specific class $c$ is defined as the harmonic mean of its precision and recall, represented by the formula:

$$F1_c = 2 \times \frac{Precision_c \times Recall_c}{Precision_c + Recall_c} \tag{2}$$

The overall Macro-F1 score is then computed as:

$$Macro\text{-}F1 = \frac{1}{|C|} \sum_{c \in \{A,U,C,G\}} F1_c \tag{3}$$

where $|C|$ represents the number of classes, namely the four types of RNA letters. This metric effectively balances the precision and recall for each letter class, providing a fair assessment of model performance, especially in cases of class imbalance.

### 3.3.3 TERTIARY STRUCTURE SELF-CONSISTENCY SCORE

To evaluate the three-dimensional structural compatibility of the designed sequence, we employ a tertiary structure prediction tool, namely RosettaFoldNA. The comparison between the design and native structures utilizes the root-mean-square deviation (RMSD) of C4' coordinates.

$$\text{RMSD}\left(x_{\text{design}}, x_{\text{pred}}\right) = \sqrt{\sum_{i=1}^{N} \frac{d_i^2}{N}} \tag{4}$$

where $d_i$ represents the distance between the $i$-th atom in the designed sequence and the corresponding atom in the predicted sequence, and $N$ is the total number of atoms being compared.

### 3.4 MODEL ARCHITECTURE

#### 3.4.1 SEQFORMER MODULE

In this study, we introduce a module named SeqFormer for processing the three-dimensional structure of RNA (Figure 1). The input to the SeqFormer module includes the local spatial orientation of C1' atoms extracted from the coarse-grained atom coordinates of RNA and the relative distances between these atoms. These inputs are defined as:

$$\text{init\_S} = \text{C1' local orientation},$$
$$\text{init\_Z} = \text{residue distance map}. \tag{5}$$

The design of the SeqFormer module is inspired by related work in protein structure prediction Jumper et al. (2021) and protein design Ren et al. (2024a), particularly the use of triangle multiplicative update and triangle attention update to satisfy constraints in three-dimensional space. In representing the three-dimensional structure of RNA, we adopt a similar approach, combining local orientation (init\_S) and residue distance map (init\_Z), and continuously updating sequence information (seq\_act) and pair information (pair\_act) through N\_recycle iterations.

In each iteration, we first fully interact the sequence dimension information and the two-dimensional information of pair, and integrate it using the outer\_product\_mean method. After integration, we use triangle multiplicative update and triangle attention update to update the pair representation. The iteration process is as follows:

---
**Algorithm 1** SeqFormer Module Iteration
---
1: **for** $i$ in range(N\_recycle) **do**
2:     seq\_act+= transition(pair\_act, agg='row')
3:     seq\_act+= transition(pair\_act, agg='col')
4:     pair\_act+= outer\_product\_mean(seq\_act)
5:     pair\_act+= triangle\_multiplication\_outgoing(pair\_act)
6:     pair\_act+= triangle\_multiplication\_incoming(pair\_act)
7:     pair\_act+= triangle\_attention\_starting\_node(pair\_act)
8:     pair\_act+= triangle\_attention\_ending\_node(pair\_act)
9:     pair\_act+= pair\_transition(pair\_act)
10: **end for**

---

#### 3.4.2 GVP MODULE

In this study, we propose a graph neural network (GNN) based method (Figure 1) to extract geometric constraint features from the Protein Data Bank (PDB). The core of our method is the construction of a graph representation $G = (S, V)$, where $S$ represents the set of scalar features and $V$ represents the set of vector features. Specifically, $S$ consists of node scalar features $node\_s$ and edge scalar features $edge\_s$, while $V$ consists of node vector features $node\_v$ and edge vector features $edge\_v$.

Firstly, we adopt the Geometric Vector Perception Graph Neural Network (GVP-GNN) Jing et al. (2020) approach to update the features of nodes and edges. The update formulas are as follows:

$$node\_s, node\_v = \text{gvp}(node\_s, node\_v)$$
$$edge\_s, edge\_v = \text{gvp}(edge\_s, edge\_v) \tag{6}$$

where gvp denotes the function used to update the features of nodes and edges.

Next, we fuse the updated node and edge features $(node\_s, node\_v, edge\_s, edge\_v)$. The message passing process can be represented as:

$$\text{message}((s_i, v_i), (s_j, v_j), edge_{ij}) \rightarrow \text{update}_{node_i} \quad \text{for} \quad j \in N_i \tag{7}$$

where $N_i$ represents the set of neighboring nodes of node $i$.

Subsequently, we concatenate the updated node features update_node with the sequence pair features seq_act, which are updated through a flow module. The concatenated features are passed through a Multi-Layer Perceptron (MLP) layer, followed by the addition of position embeddings:

$$gvp\_seq\_act = \text{mlp}(\text{concat}(update\_node, seq\_act)) \tag{8}$$

$$gvp\_seq\_act = gvp\_seq\_act + \text{position\_embedding} \tag{9}$$

Finally, we use a Transformer layer to process the concatenated features:

$$gvp\_seq\_output = \text{transformer}(gvp\_seq\_act) \tag{10}$$

To predict the likelihood of each nucleotide position having bases (A, U, C, G), we designed an MLP layer:

$$seq\_prob\_logit = \text{mlp}(gvp\_seq\_output) \tag{11}$$

By constructing GVP, we are able to extract richer geometric information. Due to the equivariance of SO(3), this method is more sensitive to the input geometric features, thereby effectively extracting unstructured geometric information. This complements the structured information of the sequence activity module for a comprehensive understanding of protein structures.

### 3.4.3 SECONDARY STRUCTURE INCLUDE

In this study, we developed a novel module named the pair constraint module for extracting secondary structure information of RNA from the Protein Data Bank (PDB) to improve the three-dimensional structure prediction of RNA. The input to this module is pair_act, which, after transformation, can predict the matching possibilities of different secondary structure positions on a seq_len × seq_len matrix.

Firstly, we extract the secondary structure of RNA from the PDB and construct a seq_len × seq_len matrix, where positions that conform to the base pairing rules {AU, CG, UG} are marked as trueHalder & Bhattacharyya (2013). Subsequently, we employ a multi-layer perceptron (MLP) to process pair_act to predict the matching possibilities at each position:

$$pair\_prob\_logit = \text{mlp}(pair\_act) \tag{12}$$

where mlp denotes a multi-layer perceptron that learns the mapping from pair_act to pair_prob_logit.

Unlike traditional RNA inverse folding models that typically focus only on one-dimensional sequence information, our pair constraint module considers the constraints of secondary structure, which aids in generating one-dimensional sequences in a more reasonable space.

### 3.4.4 LLM

In this study, we explore how to leverage the rich representational capabilities of large language models in BiRNA-BERTTahmid et al. (2024) for RNA sequences by generating diverse sequence information through a sequence module and superimposing this information onto the existing sequence activity (seq_act). Additionally, we introduce a recycling mechanism that allows the model to update errors in the next iteration process without increasing the model size.

Specifically, we first process the sequence embedding (sequence embedding) through a multi-layer perceptron (MLP), and then add the result to the sequence result from the previous iteration (r_seq_prev) to update the current sequence result (r_seq). This process can be represented by the following formula:

$$r_{\text{seq}} = r_{\text{seq}} + \text{MLP}(\text{LLMEmbed}(s)) + r_{\text{seq}_{\text{prev}}} \tag{13}$$

where LLMEmbed($s$) represents the embedding representation of the sequence $s$ by a large language model, MLP is a multi-layer perceptron that further processes the embedding, and $r_{\text{seq}_{\text{prev}}}$ represents the sequence result from the previous iteration.

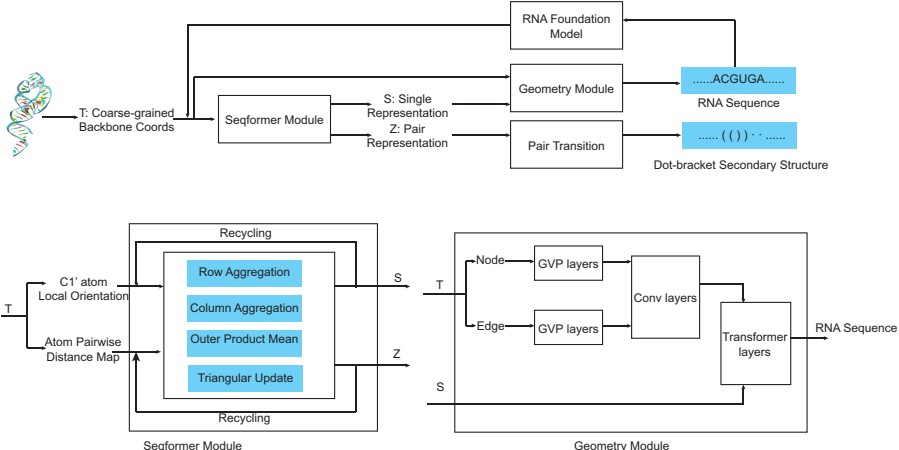

Figure 1: GeoRDe Model Architecture with SeqFormer Module and GVP Module

## 3.5 TRAINING LOSS

In this study, we propose a training loss calculation method that comprehensively considers the outputs of the sequence (seq) and pair modules. To effectively train the model, we employ the cross-entropy loss to evaluate the outputs of these two modules. Specifically, we define the total loss (Loss) as the weighted sum of the sequence module loss and the pair module loss, with weight coefficients $\alpha$ and $\beta$. This method can be represented as:

$$\text{Loss} = \alpha l_{ce}(\text{seq}_{\text{prob}}, \text{seq}) + \beta l_{ce}(\text{pair}_{\text{prob}}, \text{pair}) \tag{14}$$

where $l_{ce}(\cdot)$ represents the cross-entropy loss function, $\text{seq}_{\text{prob}}$ and seq denote the predicted output and the actual output of the sequence module, respectively, and $\text{pair}_{\text{prob}}$ and pair denote the predicted output and the actual output of the pair module, respectively.

## 4 EXPERIMENT

We conducted comparative evaluations of five distinct methodologies on four datasets, focusing on sequence recovery rate and Macro-F1 scores. Additionally, we assessed the capability of the predicted sequences in terms of three-dimensional structure prediction accuracy. These methods represent the state-of-the-art approaches for sequence design based on protein or RNA structures. All methods were trained and tested on RNA structural datasets that were meticulously divided into training, validation, and testing sets.

### 4.1 SEQUENCE DESIGN ON PRIMARY DATASETS

We initially compared the performance of these methods in nucleic acid sequence design. The testing data were categorized based on sequence length into short (less than 50 nucleotides), medium (50-100 nucleotides), and long (greater than 100 nucleotides) sequences. Both the gRNAde and RDesign datasets leverage significant collections of known RNA 3D structural data, yet they utilize different sets of RNA data. The gRNAde dataset incorporates all class member RNA structures, preserving all corresponding structures post-sequence clustering to enrich structural diversity. In contrast, the RDesign dataset employs a subset of representative RNA structures, processed to enhance dissimilarity between the test set and training data. Consequently, the RDesign dataset exhibits relatively lower performance. Training on these two datasets, our results indicate that GeoRDe

demonstrates state-of-the-art performance across both. The Performance on gRNAde dataset and Rdesign dataset are shown in Table 1 and Table 2, respectively.

Table 1: Recovery and Macro-F1 on gRNAde dataset.

| Method | Recovery(%) | | | | Macro F1(x100) | | | |
|---|---|---|---|---|---|---|---|---|
| | Short | Medium | Long | All | Short | Medium | Long | All |
| StructGNN | 0.4053 | 0.4453 | 0.4397 | 0.4312 | 0.3122 | 0.4244 | 0.4014 | 0.3293 |
| PiFold | 0.5000 | 0.5965 | 0.5711 | 0.5686 | 0.4100 | **0.5827** | **0.5229** | 0.4413 |
| RDesign | 0.4666 | 0.5676 | 0.5508 | 0.5385 | 0.3819 | 0.5611 | 0.5029 | 0.4252 |
| gRNAde | 0.4543 | 0.4939 | 0.4945 | 0.4857 | 0.4356 | 0.4772 | 0.4594 | 0.4695 |
| GeoRDe | **0.6002** | **0.7007** | **0.6695** | **0.6645** | **0.4714** | 0.5573 | 0.5092 | **0.528** |

Table 2: Recovery and Macro-F1 on RDesign dataset.

| Method | Recovery(%) | | | | Macro F1(x100) | | | |
|---|---|---|---|---|---|---|---|---|
| | Short | Medium | Long | All | Short | Medium | Long | All |
| StructGNN | 0.3182 | 0.3077 | 0.2805 | 0.3111 | 0.3407 | 0.304 | 0.2497 | 0.3024 |
| PiFold | 0.375 | 0.4676 | 0.4522 | 0.4167 | 0.3877 | 0.4494 | 0.4458 | 0.4348 |
| RDesign | 0.3777 | 0.4841 | 0.4375 | 0.4382 | 0.389 | 0.4919 | 0.4282 | 0.4433 |
| gRNAde | 0.3744 | 0.3581 | 0.3557 | 0.3755 | 0.3505 | 0.3554 | 0.3414 | 0.3603 |
| GeoRDe | **0.4932** | **0.5787** | **0.5766** | **0.5267** | **0.4675** | **0.5861** | **0.5653** | **0.5515** |

## 4.2 SEQUENCE RECOVERY RATE ON ADDITIONAL DATASETS

We assessed the performance of models trained on the gRNAde dataset using additional datasets. The CASP15Elofsson (2023) and RNA-PuzzleMagnus et al. (2020) datasets are two well-known, independent datasets. The CASP15 dataset, a comprehensive collection of RNA structures not accessible during the training phase, provides a platform for evaluating the model's capacity to generalize to novel structural data. Likewise, the RNA-Puzzle dataset offers a diverse and challenging array of RNA structures. Our findings, as illustrated in Table 3, demonstrate that GeoRDe sustains superior performance when extended to these external datasets, with sequence recovery rates that are competitive with the most advanced methods available. This indicates that GeoRDe maintains high performance when applied to these external datasets, indicating its robust generalization capabilities to new RNA structures.

Table 3: Recovery and Macro F1 on CASP15 and RNA-puzzle dataset.

| Method | Recovery(%) | | Macro F1(x100) | |
|---|---|---|---|---|
| | CASP15RNA | RNA-puzzle | CASP15RNA | RNA-puzzle |
| StructGNN | 0.4329 | 0.4486 | 0.3627 | 0.4195 |
| PiFold | 0.4262 | **0.6324** | 0.3859 | 0.6265 |
| RDesign | 0.3642 | 0.4839 | 0.3328 | 0.4515 |
| gRNAde | 0.3044 | 0.3292 | 0.2977 | 0.3286 |
| GeoRDe | **0.4623** | 0.6310 | **0.4016** | **0.6442** |

## 4.3 TERTIARY STRUCTURE RECOVERY EXAMPLES

In further assessing the performance of the GeoRDe model, we focused on the three-dimensional structural prediction accuracy of RNA sequences designed by GeoRDe in Figure 2. To this end, we selected a series of RNA sequences designed by GeoRDe and predicted their three-dimensional structures using the RosettaFoldNABaek et al. (2024) tool. We observed that the predicted structures exhibited low root-mean-square deviation (RMSD) from the original target structures, demonstrating GeoRDe's exceptional ability to preserve the structural integrity of designed sequences in three-dimensional space. These results not only substantiate GeoRDe's efficiency in sequence design but

also showcase its accuracy in structural prediction, providing a reliable tool for future RNA design and functional studies.

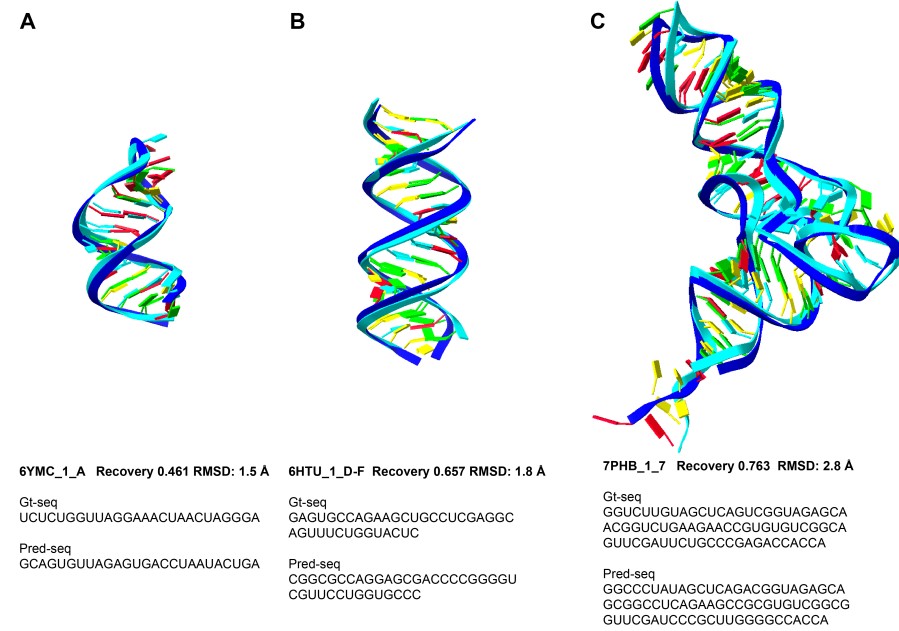

Figure 2: Visualization of GeoRDe's designed examples.

## 4.4 ABLATION STUDY

To systematically evaluate the contribution of various components within the GeoRDe model, we performed a series of ablation studies, the results of which are summarized in Table 4. As a baseline, we employed the SeqFormer module for independent RNA sequence prediction. By introducing key node and edge features and integrating the GVP module, we significantly enhanced the model's performance. This notable improvement underscores the pivotal role of three-dimensional structural information encoded by the GVP module in bolstering the accuracy of sequence design. Furthermore, incorporating RNA secondary structure information and embedding vectors from pre-trained language models marginally refines performance, achieving optimal results.

Table 4: Ablation Study

| Method | gRNAde dataset | | RDesign dataset | |
|---|---|---|---|---|
| | Recovery(%) | Macro F1(x100) | Recovery(%) | Macro F1(x100) |
| Baseline | 0.5393 | 0.4351 | 0.4316 | 0.4198 |
| Baseline + LLM | 0.5384 | 0.4357 | 0.4391 | 0.3345 |
| Baseline + GVP | 0.6565 | 0.5224 | 0.5187 | 0.5435 |
| Baseline + Secloss | 0.5506 | 0.4403 | 0.4408 | 0.4516 |
| Baseline + All | 0.6645 | 0.5280 | 0.5267 | 0.5515 |

## 5 CONCLUSION

In this study, we have presented GeoRDe, a novel algorithm designed to address the RNA inverse folding problem. Its innovative approach to handling the geometric constraints of RNA molecules, coupled with the integration of secondary structure information and evolutionary constraints.Our approach has demonstrated significant advancements in the field of RNA sequence design, offering

a comprehensive solution that not only aligns with the structural intricacies of RNA molecules but also exhibits strong generalization capabilities across different datasets. In conclusion, GeoRDe represents a significant step forward in the field of RNA sequence design, positions it as a powerful tool for both research and therapeutic development.

Despite GeoRDe's outstanding performance in multiple aspects, there are still limitations in its performance evaluation. The current metrics used, such as sequence recovery rate and F1 score, only partially reflect the accuracy of computational design. To comprehensively assess the model's performance, further experimental validation is required to ensure its reliability and accuracy in practical applications.

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
