# OpenReview forum: "GeoRDe: Tertiary Structure-based RNA Design With Multiple Geometric Constraint"
_ICLR.cc/2025/Conference — ICLR 2025 Conference Withdrawn Submission_

### Official Review · Reviewer_Uwp5 · 2024-10-20

**Soundness:** 3
**Presentation:** 1
**Contribution:** 3
**Rating:** 3
**Confidence:** 4

**Summary:**

The paper proposed an interesting RNA design method considering the geometric constraints through RNAformer and an RNA language model. It conducted extensive experiments to compare with existing approaches.  The method design and implementation are solid, the paper is easy to follow. However, the paper does not meet some presentation format for such a conference, it needs to be polished a bit.

The results are surprisingly well on both datasets, if the authors could explicitly explain their model design and experiment details, it could be a good paper to appear in ICLR 2025.

**Strengths:**

- The authors noticed a common constraint of existing graph-based structure encoders for RNA design, it is an inspiring finding in pushing this task forward.
- The experiments are solid, as this is a newly proposed field, and the datasets are not organized according to protein design. The paper took that into account and cross-compared the existing approaches.
- The seqformer seems to be a good addon to the graph modules.
- The model achieved good results (much better) compared to the current SOTA.

**Weaknesses:**

- The way to write functions and modules could be corrected! From Equations 5 to 11, it is just a model pipeline, it could be shown in a better way.
- Typo in the citation format, check with citep/citet/cite commands.
- Why do the authors use methods like pair_transition/triangle_attention… in seqformer?  How does it function?
- Incorporating secondary structure into training is not a novel thing, RDesign has done that before. Maybe a clarification of the differences would be helpful in distinguishing this paper’s novelty.

**Questions:**

- I am curious about the feature the paper used in baseline models, especially for PiFold as it is a protein design method, it should not have such good results in RNA. Did the authors replace the structural feature extraction part with their own?
- For the claim of resolving geometric constraints, it might be better to have a subsection to discuss that, as it is the main argument for its novelty.

**Details Of Ethics Concerns:**

No related concern.

---

### Official Review · Reviewer_29Vp · 2024-10-30

**Soundness:** 1
**Presentation:** 1
**Contribution:** 1
**Rating:** 1
**Confidence:** 4

**Summary:**

This work presents an RNA inverse folding model called "GEORDE" to predict RNA sequences based on RNA back-bone structures. The authors present an RNAformer module, which is a copy of AlphaFold 2's EvoFormer, applied to nucleotide sequences instead of amino acid sequences. This is input to GVP module that predicts the sequence, and an additional pairformer module predicts secondary structure.

**Strengths:**

The task of RNA tertiary structure prediction and inverse folding is notoriously hard due to the nature of the more flexible and conformationally instability of RNA structure, but also the lack of experimentally resolved structural data in the PDB.
Testing methods from protein structure design, in this case a translation of AF's evoformer is an interesting idea, however, it is not original enough to justify acceptance. Especially in light of the overall poor quality of the presentation of this work.

**Weaknesses:**

Overall this submission has substantial shortcomings in the categories of scientific writing and presentation, study design and scientific rigour, result presentation and soundness of conclusion as well as originality. I won't list all details, but I encourage the authors to review their work in light of each category. I will provide examples for each.
Scientific writing and presentation:
1. Please adhere to the citation style of ICLR, the author name is within the text and it makes it hard to read.
2. Please make use of more figures that explain the architecture, e.g. Figure 1 appears very late in the text and is sparsely referred too.
3. Organisation: The introduction and related work are a little repetitive and overblown. I'd prefer more preciseness and instead more focus on a background section that introduces for example AlphaFold 2's evoformer in mathematical detail.
4. There is a mix of writing styles: $.3.1 uses pseudocode, 3.4.2 uses a different style of pseudo-code in text. Generally a lot of notations vary, in the introduction sequence length is denoted as "N", later in the manuscript it becomes "seq_len". I would advise to not use pseud-code in text and rather use mathematical description of the operations, e.g. in line 268 " message(...) -> update" -- what does that mean?

Study design and scientific rigour:
1. Several choices are not well explained or motivated:
a) Choice nucleic acid back-bone reference. The authors chose C4', C1' and N1/N9 atoms without any reference. To my knowledge, the canonical RNA back-bone definition is defined by the phosphate bond between C3' and C5' atoms. The RNA-TM score is also defined on the C3': https://zhanggroup.org/papers/2019_24.pdf .
b) How can we assure that the combination of C4', C1' and N1/N9 angles does not leak the nucleotide identity? How was the nearest neighbour area of 12 Angstrom defined -- is there a reference or an ablations study?
c) It looks like rmsd was calculated between forward folded predicted sequence and ground truth structure. This disregards the bias of the imperfect forward folding model. It should be the RMSD or TM-score between the forward folded ground truth sequence and predicted sequence
d) the datasets are not well referenced, there is no information about size.
e) where are the PIFOLD and other model's performance coming from? Did you retrain those models?

Result presentation and soundness of conclusion:
1. Please provide std deviations or some form of statistical boundary to assess significance of differences in the results in all presented tables
2. Where do the values of all other models come from? Were they retrained on the same data?
3. How was the train/test split performed?
4. Sequence recovery on additional datasets: What is the overlap between CASP15 and your training data? Why are only the results from the gRNAde training presented? Why is there no training on both datasets (gRNAde and RDesign) presented on CASP15?
5. Tertiary structure prediction: This feels a lot like cherry picking. No statistical results are presented, but "we selected a series of RNA sequences designed by GeoRDe" -- how were those three sequences selected? In line 430: "predicted structures exhibited low RMSD" -- what does "low" mean? Again, this needs a statistical result over more than three hand-picked samples.
6. The Conclusion section reads a little overblown and I recommend the authors to tame down the claims a little bit "Our approach has demonstrated significant advancements in the field of RNA sequence design" or "[...] positions it as a powerful tool for both research and therapeutics".
Originality:
In section 4.4.1 it is claimed that SeqFormer is inspired by Jumper's work on protein design. This paper's algorithm 1 is a one-to-one copy of Jumper et al. Algorithm 6 in the AF2 supplementary (https://static-content.springer.com/esm/art%3A10.1038%2Fs41586-021-03819-2/MediaObjects/41586_2021_3819_MOESM1_ESM.pdf)

**Questions:**

See above.

---

### Official Review · Reviewer_i1aB · 2024-10-31

**Soundness:** 3
**Presentation:** 2
**Contribution:** 2
**Rating:** 3
**Confidence:** 3

**Summary:**

The paper introduces GeoRDe, a novel algorithm for RNA design focused on the inverse folding problem. By integrating a series of innovative designs to tackle geometric constraints, GeoRDe achieves significant advancements in the RNA inverse folding task.

**Strengths:**

- The performance significantly surpasses the reported baseline methods.

**Weaknesses:**

- The model lacks innovation, as most techniques have been extensively used in the protein design field, such as employing language models (ESMFold/LM-Design), GVP. The Seqformer Module also closely resembles Alphafold2.

- There is no detailed introduction to the dataset.

- The evaluation metrics in the experimental section are very limited, focusing almost exclusively on nucleotide recovery.

- There is no appendix, lacking more detailed explanations.

**Questions:**

- All the metrics are related only to nucleotide recovery; however, as an inverse-folding task, it's also crucial to determine if the predicted sequence can fold into the target structure. I'm quite puzzled by the mention of the self-consistency calculation method in section 3.3.3, yet in the experimental section, only three case scRMSDs are presented in 4.3. Why aren't the scRMSDs for all samples reported in sections 4.1 and 4.2?

- In the ablation study, it seems that the RNA language model had minimal impact. Why continue using it? Are there any strategies to better leverage the language model? Concerning the language model again, you mentioned in the summarized contributions that the RNA language model aids in enhancing evolutionary reliability, but this enhancement doesn’t seem to be reflected in the subsequent evaluations.

- Why not use generative models like diffusion/flow-match models for sequence generation, or iterative refinement?

- Besides RNA inverse-folding, what other tasks can GeoRDe perform?

---

### Official Review · Reviewer_i5Tx · 2024-11-03

**Soundness:** 2
**Presentation:** 1
**Contribution:** 2
**Rating:** 1
**Confidence:** 4

**Summary:**

While this paper has presented interesting ideas related to RNA inverse design, the current form of the presentation is far from the standard that one would expect for the main ICLR proceeding. The citation appears to be misplaced everywhere, and the math equations are not compile corrected. It is inconceivable to imagine how such visual mistakes can be overlooked before the authors submited this manuscript.

**Strengths:**

Interesting chimeric architecture as shown in Figure 1. The pipeline may work better, and even generate some useful biological hypotheses, if more care has been put into the approach.

**Weaknesses:**

- Limited novelty in machine learning approach. The main approach is a geometric graph neural network (i.e., GVP) plus something adopted from Alphafold (i.e., triangular attention), and something else from an RNA LLM.
- Upon reading the paragraph in the introduction (line 60-70), I feel that it was written by a LLM (and coincidentally, in line 70, it should "harnessing large langauage models"). In fact, a lot of places in this paper felt like it was written by AI. For example, in line 161-162, "... their relative contact distances are harnessed as pair features to enhance the characterization of the local environment surrounding each atom.". I have never seen anyone working in this field (RNA, ML) ever wrote something like this. It just feels so off.
- inconsistent naming. The module is originally introduced in the introduction as RNAformer, but in section 3.4 it becomes SeqFormer.
- Although all necessary results have been provided, the explanation from a biological viewpoint is lacking, rendering the whole result section nothing more but a benchmarking effort.

**Questions:**

- Line 48-49, RDesign didn't encode RNA 3D structures using multi-state graph neural networks. Besides, it has been discussed in gRNAde that RDesign has significant performance (and perhaps overfitting) issue. So, it may be better to discuss these two methods separately.
- Why would you ever put evaluation metrics before the actual methodology? It is so strange.
- How meaningful are the included LLM embeddings, in particular on RNA aptamers with no apparent sequence evolution? From your ablation results on Table 4, it seems the LLM improvement is neglible.

---

### Official Review · Reviewer_fmvj · 2024-11-09

**Soundness:** 2
**Presentation:** 3
**Contribution:** 2
**Rating:** 5
**Confidence:** 3

**Summary:**

This paper proposes GeoRDe, which learns the geometric constraints of RNA molecules in cooperation with geometric graph networks. To enhance the specificity of sequence generation, the authors have integrated secondary structure information as labels, ensuring that the designed sequences align more closely with secondary structure constraints. They also include RNA language models to understand average evolutionary constraints. GeoRDe has demonstrated superior performance under identical training data conditions and has also showcased generalization capabilities on the independent casp15 and RNA-puzzle datasets.

**Strengths:**

* This paper tackles RNA inverse folding problem, which is an important problem in structural biology
* The results demonstrate that the proposed method substantially outperforms the baselines

**Weaknesses:**

* The proposed method is a minor modification of previous method gRNAde, with additional features like secondary structures and language model features. While these features are important, the algorithmic innovation is rather limited.

**Questions:**

* Have you also tried incorporating multiple RNA structures? RNA structures are quite flexible so it may have multiple plausible structures.

---

### Note · Authors · 2024-11-21

I have read and agree with the venue's withdrawal policy on behalf of myself and my co-authors.